# How Does Economic Policy Uncertainty Affect Momentum Returns? Evidence from China

**Peizhi Zhao * and Yuyan Wang**

Division of Business and Management, Beijing Normal University-Hong Kong Baptist University United International College, Zhuhai 519087, China; p930004036@mail.uic.edu.cn
* Correspondence: l630013052@alumni.uic.edu.cn

**Abstract:** Economic policy uncertainty has been identified as a new macroeconomic risk factor that harms the stock market's profitability. This paper examines the impact of the Chinese EPU levels on one of the most famous financial anomalies—momentum returns. A new EPU index based on mainland China newspapers is used to obtain more accurate EPU–momentum relations. We selected 3958 Chinese listed companies' stocks from 2011 to 2022 to establish time-series (TSM) and returns signal momentum strategies (RSM). Although the momentum effect in the Chinese stock market is weak, the EPU-based dynamic-threshold RSM strategies yield significant positive excess returns: eight times more excess returns than conventional fixed-threshold strategies. We used the ordinary least squares regression model (OLS), and the event study method only identified robust negative EPU–momentum relationships in the Chinese stock market during high-EPU stages. Surprisingly, the negative relationship between EPU and momentum returns turns positive during expansion cycles. We explain this phenomenon as follows: expansions increase Chinese investors' confidence, and uncertainties reduce market manipulations.

**Keywords:** momentum returns; economic policy uncertainty; event study method; emerging market

## 1. Introduction

The novel Economic Policy Uncertainty (EPU) index quantifies a country's level of economic policy uncertainty according to the newspaper archives of each country. It counts the occurrence frequency of keywords relevant to economic and policy uncertainty to determine the magnitude of a country's uncertainty from an economic policies perspective (Baker et al. 2016). EPU reflects the uncertainty degree of regulatory, monetary, and fiscal policy in a country (Phan et al. 2018) and significantly affects financial markets' returns (Gu et al. 2021). The impact of EPU on financial markets is usually negative, and it substantially intensifies stock markets' volatility and grants investors' difficulties in earning excess returns from stock investments (Liu and Zhang 2015). Phan et al. (2018) tested the relationship between EPU and stock market excess returns and obtained the same negative conclusion in 16 countries. While higher EPU brings lower stock returns, investors unremittingly seek opportunities to achieve higher returns through different trading strategies.

The momentum effect is one of the best well-known financial anomalies (Blitz et al. 2020). Based on the momentum effect, as one of the most frequently used stock trading strategies, momentum strategies have long been used in trading stocks and have achieved considerable excess returns worldwide (Antoniou et al. 2007). However, the effectiveness of momentum trading strategies differs in various markets. In the Chinese stock market, either the cross-sectional momentum strategies (XSM) or time-series momentum strategies (TSM) yield lower excess returns compared to the US stock market (Cheema and Nartea 2017). In their further study, Cheema et al. (2020) suggested that the momentum effect no longer

exists in the Chinese stock market due to unknowable reasons other than time-varying investment in risky assets. Gu et al. (2021) claimed that investors are unwilling to apply momentum trading strategies when exposed to high EPU environments, since such uncertainties will impede their judgment of momentum continuity. Evidence shows that high-level EPU situations will weaken the momentum effect (Paule-Vianez et al. 2021). In the U.S. stock market, momentum strategies yield 1.11% less when the EPU index rises one standard deviation (Gu et al. 2021). The implication is that the excessive EPU level would completely erase the momentum return from the stock markets. However, Cheema and Nartea (2014) claim that the uncertainty level will positively promote momentum returns in China. Due to Chinese investors' low level of individualism, uncertainties will motivate their overconfidence and strengthen the momentum returns. In this case, as one of the representatives of emerging markets, the Chinese stock market may perform differently than developed markets. The existing literature has not researched the influence of the EPU index on the momentum returns from the emerging stock market. It is vital to study the EPU–momentum returns relationships comprehensively and find potentially profitable strategies. Hence, this paper raises these questions: How do different levels of EPU influence momentum returns from the Chinese stock market? How do investors make profits from momentum strategies under different EPU levels?

This paper aims to investigate the relationship between the Chinese EPU index and momentum returns from the Chinese stock market through OLS regressions and event study method and figure out possible profitable momentum trading strategies in emerging markets for investors under the influence of different EPU magnitudes. The regression analysis also concerns other macroeconomic factors to ensure the reliable EPU–momentum return relationships. Apart from the conventional time-series momentum strategies (TSM) (Moskowitz et al. 2012), this paper adopts the return signal momentum (RSM) proposed by Papailias et al. (2021) as an alternative momentum strategy, since they asserted that their RSM perform better than traditional TSM strategies in excess returns and risk resistance. This paper will compare the performance of these two momentum strategies and suggest to investors possible alternative profitable momentum strategies. In light of offsetting the possible biases due to the relatively short time span of available data, this paper applies an event study method to obtain robust relationships between EPU and stock momentum returns.

The findings of this paper are as follows: (1) We identified weak momentum effects in the Chinese stock market, as the existing literature shows. (2) We adopted fixed threshold RSM strategies and EPU-based dynamic threshold RSM strategies compared to conventional TSM strategies. RSM strategies yield higher momentum excess returns than TSM strategies, while the dynamic threshold RSM strategies have eight times higher profits than TSM strategies. (3) The negative influences of EPU on RSM returns are less critical than on TSM returns.

Further, we summarize the contributions of this paper as follows. This paper has filled the knowledge gap, in that no existing literature has studied the relationship between EPU and momentum returns in emerging markets. In the Chinese market, high-EPU situations will have the same negative impacts on momentum returns that the previous literature has identified in the U.S. market, but we obtained a positive EPU–momentum returns relationship during the medium-EPU stages. The EPU-based dynamic threshold RSM strategies can effectively remove EPU's negative impacts. Investors could make considerable excess returns from momentum strategies and provide supportive empirical evidence for investors and scholars by using dynamic threshold momentum strategies. Moreover, we have identified that the EPU has opposite effects on momentum returns during expansions, which provides investors with potential opportunities for higher profits. Second, to the best of our knowledge, this paper is the first one that employs the event study methods to research the relationship between EPU and momentum returns. The event study analysis of high-EPU events yields robust results as regression analyses. It

precisely shows the influence of each high-EPU event on each month's momentum returns at 36 months.

The remainder of this paper is planned as follows. Section 2 conducts the EPU and momentum strategy-related literature review. Section 3 describes the data and intended research methodology. Section 4 is the presentation of regression results and explanations of implications. Section 5 investigates the relationship between the Chinese EPU and momentum returns while controlling the macroeconomic factors. Section 6 shows the results of the event study analysis. Section 7 discusses the results obtained from Sections 5 and 6. Section 8 concludes.

## 2. Literature Review

Scholars have verified that the EPU index is a robust measure of a country's level of economic policy uncertainty (Perić and Sorić 2018; Ghirelli et al. 2019; Lyke 2020). Notably, Huang and Luk (2020) questioned the accuracy of the existing Chinese EPU index since it is derived merely from Hong Kong newspaper information. The new Chinese EPU index was generated according to the weighted average of the normalized frequency of EPU-related keywords appearing in 134 mainland China newspapers, reflecting a more accurate economic policy uncertainty level in mainland China. Such comparison suggests scholars should use the new EPU index when studying the Shanghai and Shenzhen stock markets to obtain more accurate results.

The existing literature indicates that EPU significantly negatively predicts the stock returns (Gong et al. 2022). Their results are consistent among 23 developed and emerging stock markets globally. The negative effects of EPU spillover on stock returns seem weaker than that in developed markets. Luo and Zhang (2020) supposed a more severe impact of the high level of EPU would lead to a share price crash at the market level. When the EPU level increases, investors seek to demand additional returns from the stocks they hold due to the high EPU levels intensifying the volatility exposed by companies. Therefore, momentum traders will demand higher excess returns during high EPU periods. However, the increase in the EPU level seems only to increase the stock market volatility contemporaneously (Kundu and Paul 2022). As the investors claim a higher 'uncertainty premium', the volatility will tend to flatten out, and therefore EPU will have slight lagging impacts on stock returns.

As for the effect of EPU on stock momentum returns, Brogaard and Detzel (2015) indicates that a one-standard-deviation increase in the US EPU index will result in 1.5% higher momentum excess returns from the U.S. stock market in the subsequent two to three months. However, EPU has a significant negative impact on contemporary stock excess returns. Gu et al. (2021) find a negative relationship between the economic policy uncertainty index and the U.S. stock market's momentum effect. They reconfirmed that a country's high EPU situations erode the momentum profits through linear regression models controlling other determinants. However, they used cross-sectional momentum excess return as a proxy of momentum effect without considering time-series momentum portfolio strategies. Moskowitz et al. (2012) identify that time-series momentum effects partially correlate with cross-sectional momentum effects. We cannot assure that EPU hurts momentum effects without considering the time-series momentum effect. Moreover, this conclusion may be biased in markets that only show the cross-sectional momentum effect. Most of the literature failed to find cross-sectional momentum effects in the Chinese stock market (Li et al. 2010; Wu 2011), while they verified the weak existence of time-series momentum effects (Shi and Zhou 2017). To evaluate the impact of EPU on stock momentum returns in the Chinese stock market, using cross-sectional momentum strategies may not be a reasonable concern.

Other than conventional time-series momentum strategies, return signal momentum strategies are sub-branches of time-series momentum strategies that relate to the time-series momentum (Papailias et al. 2021). RSM depends on the sign of past returns, reduces the impact of volatility, and yields higher profits than TSM strategies. Moreover, the ranking

period's threshold selection is more flexible than TSM and XSM strategies in achieving higher momentum profit. The flexible threshold selection of RSM also has a significant deficiency because it needs a relatively more extended ranking period. Investors cannot allocate threshold flexibly on the ranking period within nine months every month.

*Business Cycle, EPU, and Momentum Returns*

Antoniou et al. (2007) find a weak explanatory ability of idiosyncratic characteristics of stock returns to stock market momentum; a financial market anomaly. However, the business cycle can better explain the momentum profitability since it may contain risk factors that potentially affect momentum returns. Recent studies agree that the EPU level will have different impacts on stock market returns depending on the business cycle (Paule-Vianez et al. 2020). The impact of macroeconomic factors on momentum returns could vary due to different business cycle stages and the idiosyncratic national economic environment. Wei (2009) suggests that the U.S. equity returns present a more significant negative response to inflation during recession periods of business cycles than in expansions. Moreover, even inflation indicates a positive impact on stock premium in European stock markets during economic expansions. In contrast, the negative influence of inflation on equity premium remains the same as in the U.S. market (Alqaralleh 2020). In developing stock markets, recession phases seem to have a more substantial negative impact on the relationship between inflation and stock returns (Cifter 2015).

Like other macro determinants, EPU shows different effects during expansion and recession periods by controlling the business cycle dummy. Adjei and Adjei (2017) claim that economic policy uncertainty negatively impacts stock return premium only when the economic environment suffers recessions. One possible explanation is that economic policy uncertainty will cause a more intensely negative impact on unemployment during economic recession periods (Caggiano et al. 2017). A higher unemployment rate will further intensify its negative impact on stock returns (Li and Matti 2020). The business cycle will significantly impact the relationship between Chinese EPU and the stock market's momentum returns. This paper will investigate the effects of other selected macroeconomic controlling factors on the EPU–momentum return relationship to obtain robust results.

## 3. Data and Methods

Distinct from the existing literature, this paper adopts the new Chinese EPU index calculated by Huang and Luk (2020) instead of Baker et al.'s Chinese EPU index to evaluate the influence of the high EPU situation on momentum returns in the Chinese stock market more accurately. Baker et al.'s Chinese EPU index merely counted the economic policy uncertainty-related keywords from the top 10 Hong Kong newspapers. Huang and Luk (2020) gathered information from 134 mainland China newspapers. The monthly Chinese EPU index data from 2011 to 2022 is obtained from the website[1] established by Huang and Luk and a total of 156 months is covered.

We selected 3958 Chinese A-share stocks listed on Shenzhen Stock Exchange and Shanghai Stock Exchange with the same time range as the EPU index. The Fama–French three factor data of the Shenzhen Stock Exchange and Shanghai Stock Exchange are obtained from the RESSET database, and we took a market-value weighted average of the data from the two Chinese stock exchanges.

We first run a linear regression to determine a relation between EPU and risk-adjusted momentum returns, respectively, as shown in Equations (1) and (2).

$$R_t = \alpha + \beta_1 EPU_{t-1} + \varepsilon_t \tag{1}$$

$$R_t = \alpha + \beta_1 EPU_{t-1} + \beta_2 MKT_t + \beta_3 SMB_t + \beta_4 HML_t + \varepsilon_t \tag{2}$$

where $R_t$ represents the monthly average excess return from momentum strategies at month $t$. $EPU_{t-1}$ represents the previous month's EPU index of month $t$. $MKT_t$, $SMB_t$, and

$HML_t$ represent the expected market excess return, size premium of small minus big, and value premium of high minus low at month $t$, respectively.

Later, we divide EPU into three levels: high, medium, and low stages, to explore the influence of different EPU levels on Chinese stock market momentum as Equation (3).

$$R_t = \alpha + \beta_{1,s}EPU_{t-1} + \beta_2 MKT_t + \beta_3 SMB_t + \beta_4 HML_t + \varepsilon_t \tag{3}$$

where $R_t$ denotes the excess return yield from momentum strategies. $s$ denotes the stages of EPU level, including high, medium, and low levels. MKT, SMB, and HML are the Fama–French three factors.

In Equation (4), we control other macroeconomic factors to study further and obtain a robust relationship between momentum excess returns and EPU in the Chinese stock market.

$$R_{t+1} = \alpha + \beta_1 EPU_t + \beta_i FF3_{t+1} + \beta_J Macro_{t+1} + \varepsilon_{t+1} \tag{4}$$

where $FF3_{t+1}$ denotes the Fama–French three factors at time $t + 1$. $Macro_{t+1}$ indicates the macroeconomic factors shown in Table 1. $i$ and $j$ represent the $i$th Fama–French factors and the $j$th macroeconomic factor, respectively. Table 1 below shows the abbreviations and detailed definitions of selected macroeconomic factors.

**Table 1.** Definitions and explanations of macroeconomic control factors.

| Macro Factors | Abbreviations | Definitions |
|---|---|---|
| Term structure | TERM | Difference between 10-year Chinese government bond and 3-month interest rate |
| Short term interest rates | STIR | 3-month interest rate divided by its 12-month moving average (Hutchinson and O'Brien 2020) |
| Market returns | MKT | Value-weighted monthly premium of CSI 300 |
| Money supply | MS | The amount of money in Chinese economy (M2) |
| Unemployment rate | UEMP | Chinese monthly unemployment rate |
| Dividend yield | DIV | Aggregate dividend yield on CSI 300 |
| Industrial production growth rate | IPG | Chinese monthly industrial production growth rate |
| Inflation | INF | Monthly inflation based on Chinese CPI |

Then we divide our samples into two pools: expansions and recessions, according to the Chinese business cycle, to study the moderating effect of the business cycle on the EPU–momentum relationship. Table 2 shows the business cycles of the Chinese economy.

**Table 2.** Business cycles in the Chinese economy. We follow the method used by Li et al. (2021) (Data from the Wind Database). We adopt seasonal adjustments to real GDP data of China from 2011 to 2021 through the X-12 procedure and use the HP filter (Hodrick and Prescott 1997) to extract the cyclical element of the Chinese GDP series. Then we apply the Bry–Boschan algorithm (Bry and Boschan 1971) to determine the turning point of Chinese business cycles.

| Period | Business Cycle |
|---|---|
| 2011Q1–2011Q3 | Recession |
| 2011Q4–2013Q1 | Expansion |
| 2013Q2–2013Q4 | Recession |
| 2014Q1–2015Q4 | Expansion |
| 2016Q1–2017Q4 | Recession |
| 2018Q1–2019Q4 | Expansion |
| 2020Q1–2021Q1 | Recession |
| 2021Q2–2021Q4 | Expansion |

The innovation in methodology in this part of this study is that we applied an event study analysis to the influence of high-EPU events on momentum returns.

The event study method benefits our paper in ensuring the robustness of our findings, and can provide a dynamic view on a monthly basis of the EPU–momentum returns relationship. Moreover, we can intuitively observe the high-EPU events' effects on momentum

returns from a visualized figure. The estimation window ($T_0$–$T_1$) is set to 60 months, the pre-event window is set to 12 months (0–$T_1$), and the post-event window is set to 24 months (0–$T_2$), as shown in Figure 1. This study defines the high-EPU events as the top 20 percentile of the Chinese EPU index from 2011 to 2022. The detailed steps and equations of implementing the event study method is shown in Appendix A.

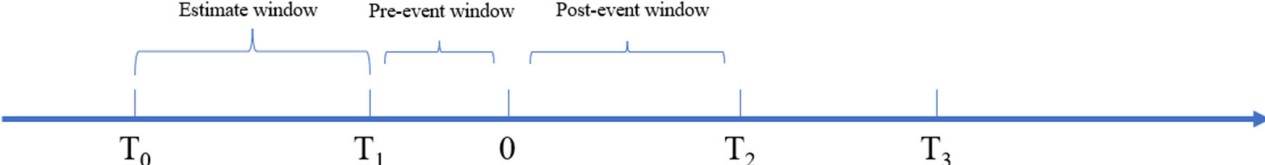

**Figure 1.** Event study method.

### 3.1. Descriptive Statistics

### 3.1.1. Fixed Threshold RSM and TSM Momentum Strategies

This section establishes time-series momentum portfolios and returns signal momentum portfolios to obtain the excess return series in order to measure the time-series momentum effect and return signal momentum effect in the Chinese stock market. To investigate the relationship between the stock market momentum effect and EPU in China, we test excess returns from TSM strategies and RSM strategies with ranking periods R = 9 and 12 months. The relative longer ranking period allows us to adjust the threshold of RSM strategies more flexibly. Following Gu et al. (2021), we skip one month between the look-back and holding periods to eliminate the short-term reversal effect.

For time-series momentum strategies, we long the winner portfolio consisting of stock with a positive cumulated past return during the ranking period and short the loser portfolio with a negative cumulated past return over the past ranking months. For fixed-threshold return signal momentum strategies, we simply set the threshold as 0.3. We long the stocks with positive returns on at least 30% months of the look-back period and short the rest of the stocks. We obtain their monthly average equally-valued excess returns for all momentum strategies and Fama–French risk-adjusted excess returns.

Table 3 shows the t-statistics of excess returns and Fama–French three-factor risk-adjusted return obtained from the RSM0.3 and TSM strategies. The Newey and West (1987) prewhitening heteroscedasticity-autocorrelation consistent (HAC) method is used to obtain the most accurate t-statistics, since the parameterized prewhitening HAC method can reduce the finite sample bias (Andrews and Monahan 1992). Most momentum strategies we established indicate an insignificant excess return. However, the RSM strategy with a 12-month (RSM 12) ranking period presents a 0.57% (t-stat = 2.04) monthly average excess return. The winner portfolio shows a 0.61% (t-stat = 1.91) average excess return per month, and the loser portfolio suggests a 0.04% (t-stat = 0.92) excess return per month. In comparison, only the winner portfolio of the TSM 12 strategy shows a significant monthly average excess return of 0.42% (t-stat = 1.72). RA Alpha represents the risk-adjusted alpha obtained through the Fama–French three factor model. All fixed threshold RSM strategies and TSM strategies show a meagre risk-adjusted alpha. The winner-minus-loser (WML) portfolio for RSM 9, TSM 9, RSM 12, and TSM 12 strategies are 0.07% (t-stat = 0.38), 0.02% (t-stat = 0.09), 0.07% (t-stat = 0.65), and 0.03% (t-stat = 0.11), respectively.

**Table 3.** Descriptive statistics of conventional static momentum and economic policy uncertainty. ** denotes 0.05 significance level, and * represents significant at 0.1 level, respectively.

| | | RSM9 | | TSM9 | | RSM12 | | TSM12 | |
|---|---|---|---|---|---|---|---|---|---|
| | | Excess Returns | RA Alphas | Excess Returns | RA Alphas | Excess Returns | RA Alphas | Excess Returns | RA Alphas |
| Winner | Return | 0.37 | 0.01 | 0.34 | −0.03 | 0.56 * | 0.01 | 0.42 * | −0.03 |
| | t-stat | (1.59) | (0.01) | (1.41) | (−0.20) | (1.91) | (0.01) | (1.72) | (−0.19) |
| Loser | Return | 0.19 | −0.07 | 0.22 | −0.05 | 0.04 | −0.08 | 0.23 | −0.06 |
| | t-stat | (0.80) | (−0.53) | (0.84) | (−0.29) | (0.92) | (−0.52) | (0.87) | (−0.32) |
| WML | Return | 0.18 | 0.07 | 0.12 | 0.02 | 0.57 ** | 0.07 | 0.20 | 0.03 |
| | t-stat | (0.75) | (0.38) | (0.41) | (0.09) | (2.04) | (0.65) | (0.72) | (0.11) |

3.1.2. Time-Varying Return Signal Momentum Strategies

Papailias et al. (2021) suggest using a dynamic threshold instead of a fixed threshold to improve the profitability of return signal momentum strategies. We set the time-varying threshold for RSM strategies according to the percentile of the Chinese EPU index, since we assume EPU will negatively affect stock momentum returns. The higher the EPU index shows at month t, the higher the threshold for winner portfolios. Table 4 shows the performance of dynamic RSM strategies with the ranking period of nine and twelve months.

**Table 4.** Descriptive statistics of dynamic threshold return signal momentum and economic policy uncertainty. ** denotes 0.05 significance level, and * represents significant at 0.1 level, respectively.

| | | Dynamic RSM 9 | | Dynamic RSM 12 | |
|---|---|---|---|---|---|
| | | Excess Returns | RA Alphas | Excess Returns | RA Alphas |
| Winner | Return | 0.66 * | 0.28 | 0.68 * | 0.18 |
| | t-stat | (1.88) | (1.47) | (1.84) | (0.98) |
| Loser | Return | −0.11 | −0.35 * | −0.03 | −0.26 |
| | t-stat | (−0.53) | (−1.95) | (−0.17) | (−1.48) |
| WML | Return | 0.78 ** | 0.62 * | 0.72 * | 0.45 |
| | t-stat | (1.98) | (1.93) | (1.80) | (1.28) |

For a ranking period of 9 months' return signal momentum strategy, the monthly average excess return is 0.78% (t-stat = 1.98), the winner portfolios gains 0.66% (t-stat = 1.88) per month on average while loser portfolios indicates a −0.11% (t-stat = −0.53) on average monthly excess return. The monthly average excess returns from the dynamic return signal momentum strategy with a 12-month ranking period yield similar excess return to the 9-month ranking period RSM strategy. The winner-minus-loser (WML) portfolio with a 12-month ranking period achieves a 0.72% (t-stat) monthly excess return. The winner portfolio and loser portfolio obtain average monthly excess return of 0.68% (t-stat = 1.84) and −0.03% (t-stat = −0.17). Interestingly, the risk-adjusted result through the Fama–French three factor model of the dynamic RSM 9 strategy shows a 0.62% (t-stat = 1.93) monthly average excess return, which is significant at a 10% level. However, the alpha value from the dynamic RSM 12 strategy still shows an insignificant 0.45% (t-stat = 1.28) monthly average excess return.

**4. Empirical Results from Regressions**

In this section, this paper starts with linear regressions on excess returns series with Fama–French three factors and EPU index to illustrate the relationships for momentum strategies. Table 5 shows the 1-month lagged effect of the EPU index on momentum returns.

**Table 5.** Month lagged effect of Chinese EPU index on momentum strategies in the Chinese stock market. *** denotes significant at 0.01 level, ** denotes 0.05 significance level, and * represents significant at 0.1 level, respectively.

|  | RSM9 | TSM9 | RSM12 | TSM12 | DRSM9 | DRSM12 |
|---|---|---|---|---|---|---|
| Winner | −0.010 ** | −0.011 * | −0.004 | −0.018 ** | 0.003 | −0.001 |
| t-stat | (−1.980) | (−1.947) | (−1.072) | (−2.148) | (0.322) | (−0.093) |
| Loser | 0.017 ** | 0.019 * | 0.012 ** | 0.026 ** | 0.005 | 0.009 |
| t-stat | (1.766) | (1.713) | (2.009) | (2.187) | (0.497) | (0.958) |
| WML | −0.027 ** | −0.030 * | −0.016 ** | −0.045 *** | −0.002 | −0.010 |
| t-stat | (−2.124) | (−1.855) | (−2.343) | (−2.651) | (−0.718) | (−0.503) |

From Table 5, this paper observes that the Chinese EPU index significantly negatively affects static TSM and RSM strategies' WML portfolios' returns. While the EPU level hurts the winner portfolios' profits, it has significant booster effects on loser portfolios' performance. Besides, this paper finds that the EPU level will significantly influence dynamic momentum strategies' returns.

*High, Medium, Low EPU Levels and Momentum Returns*

From our preliminary results, investors may choose these strategies in high EPU stages to hedge against Chinese economic policy uncertainty risks. To further illustrate this assumption, we divide the Chinese EPU index into three groups, scilicet high, medium, and low, to investigate the EPU–momentum return relationships. The regression results are shown in Table 6.

**Table 6.** Effect of high, medium, and low EPU stages on momentum returns in the Chinese stock market. *** denotes significant at 0.01 level, ** denotes 0.05 significance level, and * represents significant at 0.1 level, respectively.

|  | High EPU | | | Medium EPU | | | Low EPU | | |
|---|---|---|---|---|---|---|---|---|---|
|  | RSM9 | TSM9 | DRSM9 | RSM9 | TSM9 | DRSM9 | RSM9 | TSM9 | DRSM9 |
| Winner | −0.042 | −0.037 | −0.065 * | 0.073 | 0.090 | 0.006 | 0.034 | 0.046 | 0.074 |
| t-stat | (−1.225) | (−1.112) | (−1.797) | (1.185) | (1.433) | (0.081) | (0.715) | (0.882) | (1.266) |
| Loser | −0.034 | −0.040 | −0.009 | −0.007 | −0.024 | 0.060 | 0.045 | 0.034 | 0.005 |
| t-stat | (−0.830) | (−0.840) | (−0.219) | (−0.132) | (−0.414) | (1.001) | (1.450) | (1.022) | (0.249) |
| WML | −0.009 | −0.003 | −0.056 | 0.079 ** | 0.114 ** | −0.054 | −0.011 | 0.012 | 0.069 |
| t-stat | (−0.285) | (−0.074) | (−1.591) | (2.076) | (2.398) | (−0.617) | (−0.220) | (0.200) | (1.152) |
|  | **High EPU** | | | **Medium EPU** | | | **Low EPU** | | |
|  | RSM12 | TSM12 | DRSM12 | RSM12 | TSM12 | DRSM12 | RSM12 | TSM12 | DRSM12 |
| Winner | −0.054 | −0.040 | −0.074 | 0.071 | 0.091 | 0.011 | 0.055 | 0.040 | 0.074 |
| t-stat | (−1.023) | (−1.241) | (−1.848) | (0.823) | (1.407) | (0.141) | (0.979) | (0.722) | (1.246) |
| Loser | −0.008 | −0.022 | 0.013 | −0.005 | −0.025 | 0.055 | 0.024 * | 0.040 | 0.005 |
| t-stat | (−0.396) | (−0.487) | (0.359) | (−0.218) | (−0.463) | (0.922) | (1.792) | (1.481) | (0.256) |
| WML | −0.047 | −0.018 | −0.087 *** | 0.076 | 0.116 ** | −0.044 | 0.031 | −0.004 | 0.069 |
| t-stat | (−1.180) | (−0.472) | (−2.982) | (1.067) | (2.040) | (−0.479) | (0.605) | (−0.006) | (1.111) |

As shown in Table 6, our results are opposite Gu et al. (2021). From Table 6, this paper observes that the high EPU level insignificantly jeopardized all WML portfolios and reduced their excess returns in the Chinese stock market. During medium EPU level periods (normal times), the economic policy uncertainty will positively influence the momentum returns. This paper did not find any influence of EPU level on the momentum returns in the Chinese stock market.

## 5. EPU–Momentum Relationships and Macroeconomic Factors

This section controls macroeconomic factors potentially influencing the relationship between the Chinese EPU index and stock momentum returns. Each momentum strategy we used has a different ranking period, and we calculate an arithmetic average of them to obtain new momentum return series for each strategy. Then, we pool the selected macroeconomic factors and the Fama–French three factor model together as control variables. Such regression allows us to simulate the Chinese macro environment as much as possible. After that, we investigate the effect of every single macroeconomic factor on the Chinese EPU–Momentum relationship to obtain robust results.

Table 7 shows the correlation coefficients among momentum returns, Fama–French three factors, and macroeconomic factors. The correlation coefficients of RSM and TSM in terms of EPU are −0.22 and −0.22, respectively. While momentum returns are significantly correlated with EPU at a 0.05 significance level, the cross-sectional correlation matrix suggests a solid statistical foundation for further analyses.

**Table 7.** Correlation analysis matrix of momentum returns, Fama–French three factors, and macroeconomic factors. *** denotes significant at 0.01 level, ** denotes 0.05 significance level, and * represents significant at 0.1 level, respectively.

| | RSM | TSM | EPU | HML | RMRF | SMB | TERM | MKT | M2 | UEMP | DIV | IPG | INF |
|---|---|---|---|---|---|---|---|---|---|---|---|---|---|
| RSM | 1.00 | | | | | | | | | | | | |
| TSM | 0.81 *** | 1.00 | | | | | | | | | | | |
| EPU | −0.22 ** | −0.22 ** | 1.00 | | | | | | | | | | |
| HML | −0.22 ** | −0.11 | 0.02 | 1.00 | | | | | | | | | |
| RMRF | 0.61 *** | 0.20 ** | −0.24 *** | −0.19 ** | 1.00 | | | | | | | | |
| SMB | 0.47 *** | 0.19 ** | −0.08 | −0.54 *** | 0.27 *** | 1.00 | | | | | | | |
| TERM | −0.10 *** | −0.04 | −0.39 *** | 0.03 | −0.08 | −0.14 | 1.00 | | | | | | |
| MKT | 0.44 *** | 0.08 | −0.02 | −0.02 | 0.94 *** | −0.00 | −0.05 | 1.00 | | | | | |
| M2 | −0.05 | −0.09 | −0.06 | −0.06 | 0.02 | −0.12 | 0.66 *** | 0.05 | 1.00 | | | | |
| UEMP | −0.04 | −0.10 | −0.13 | −0.13 | 0.03 | −0.09 | 0.32 *** | 0.04 | 0.80 *** | 1.00 | | | |
| DIV | −0.03 | 0.02 | 0.05 | 0.05 | −0.12 | 0.12 | −0.22 ** | −0.18 ** | −0.45 *** | −0.24 *** | 1.00 | | |
| IPG | −0.07 | −0.07 | 0.09 | 0.09 | −0.06 | 0.01 | −0.05 | −0.05 | 0.02 | 0.04 | 0.03 | 1.00 | |
| INF | −0.03 | −0.09 | −0.11 | −0.11 | 0.01 | 0.09 | −0.11 | 0.01 | −0.03 | 0.16 * | 0.02 | −0.04 | 1.00 |

Furthermore, we set up VIF analysis to examine the potential existence of multicollinearity among selected macroeconomic factors as shown in Table 8. Table 8 shows that all variables' centered VIF value for RSM and TSM returns do not exceed ten. It suggests that the degree of multicollinearity among selected macroeconomic risk factors are slight to be ignored, and does not significantly affect coefficient estimates.

**Table 8.** Centered VIF for RSM and TSM returns with selected macroeconomic variables.

| Variable | Centered VIF for RSM | Centered VIF for TSM |
|---|---|---|
| EPU | 2.69 | 2.79 |
| HML | 5.90 | 4.02 |
| RMRF | 5.35 | 7.26 |
| SMB | 5.59 | 4.47 |
| TERM | 5.16 | 7.78 |
| STIR | 2.34 | 3.16 |
| MKT | 9.41 | 5.34 |
| M2 | 8.99 | 7.47 |
| UEMP | 6.93 | 7.37 |
| DIV | 3.40 | 3.98 |
| IGP | 8.90 | 4.50 |
| INF | 2.02 | 1.77 |

The left panel of Table 9 shows the result of full sample regressions. We find a significant negative impact of the Chinese EPU level on the winner portfolio excess returns of TSM strategies at a 0.05 significance level. Although EPU negatively impacts RSM returns, such a relationship is insignificant. The consistent negative association between

EPU level and momentum returns allows us to support the existing literature on the negative influence of EPU on momentum returns.

**Table 9.** Effect of EPU on momentum returns in the Chinese stock market. *Note.* The left panel shows the full sample results while the medium and right panels indicate the EPU–momentum relationship in the Chinese stock market during expansion and recession periods, respectively. *** denotes significant at 0.01 level, ** denotes 0.05 significance level, and * represents significant at 0.1 level, respectively.

| | Full Sample | | Expansions | | Recessions | |
|---|---|---|---|---|---|---|
| | **RSM** | **TSM** | **RSM** | **TSM** | **RSM** | **TSM** |
| $EPU_{t-1}$ | −0.005 | −0.023 ** | 0.012 * | 0.003 | 0.001 | −0.003 |
| | (−1.049) | (−2.267) | (1.839) | (0.253) | (0.103) | (−0.029) |
| HML | 0.090 | 0.026 | 0.125 | 0.037 | 0.176 ** | 0.208 |
| | (1.256) | (0.156) | (0.862) | (0.375) | (2.144) | (1.634) |
| RMRF | 0.773 *** | 0.959 *** | 0.408 * | 0.568 * | −0.748 | −0.758 |
| | (10.067) | (5.482) | (1.750) | (1.758) | (−1.362) | (−0.959) |
| SMB | 0.026 | −0.230 | −0.061 | −0.235 ** | 0.401 | 0.378 *** |
| | (0.274) | (−1.168) | (−0.774) | (−2.131) | (1.434) | (4.318) |
| $TERM_{t-1}$ | −0.220 | −0.261 | 0.934 | 0.021 | 0.896 | 1.911 |
| | (−0.768) | (−0.596) | (1.627) | (0.030) | (0.585) | (1.112) |
| $STIR_{t-1}$ | 0.001 | −0.002 | −0.059 | −0.088 | 0.057 | 0.088 |
| | (0.101) | (−0.089) | (−0.969) | (−1.289) | (1.651) | (0.004) |
| $MKT_{t-1}$ | −0.054 *** | −0.085 *** | −0.034 * | −0.056 ** | 0.077 * | 0.065 |
| | (−5.585) | (−3.408) | (−1.780) | (−2.397) | (1.699) | (0.830) |
| $MS_{t-1}$ | −0.197 | −0.883 | −1.841 * | 0.100 | 3.541 | −1.873 |
| | (−0.387) | (−0.789) | (−1.753) | (0.062) | (1.653) | (−0.416) |
| $UEMP_{t-1}$ | 0.009 | −0.045 | 0.754 | 0.251 | 0.006 | 2.598 ** |
| | (0.239) | (−0.086) | (0.391) | (0.104) | (0.006) | (2.175) |
| $DIV_{t-1}$ | −0.051 * | −0.080 | −0.063 | −0.042 | 0.377 | 0.153 |
| | (−1.673) | (−1.197) | (−1.295) | (−1.035) | (1.487) | (0.762) |
| $IPG_{t-1}$ | −0.030 ** | −0.031 | −1.547 | 0.781 | 0.002 | −0.002 |
| | (−2.211) | (−1.006) | (−1.221) | (0.495) | (0.068) | (−0.101) |
| $INF_{t-1}$ | −0.115 | −0.225 | 0.569 | 0.594 | −0.399 ** | −1.039 |
| | (−0.859) | (−0.880) | (0.800) | (0.918) | (−2.069) | (−0.807) |

Furthermore, the medium and right panels control the business cycle dummy of the Chinese economy. The business cycle will significantly influence the EPU–momentum return relationship. During the expansion periods of the Chinese economy, the EPU level will increase the momentum excess returns, and such a positive impact is significant at the 0.1 significance level on RSM returns. However, the recession stages will have a neutral effect on the momentum returns.

Table 10 shows the effect of EPU on RSM strategies returns under the control of each macroeconomic factor. Whether we control individual macroeconomic factors or control individual macroeconomic factors with Fama–French three factor risk-adjusted momentum returns, the negative impacts of EPU on the Chinese stock market's return signal momentum returns are constant. Although the significance of the level of the relationship of the EPU–momentum returns seems insignificant, they are very close to a 0.1 significance level. Additionally, Table 11 shows the effect of EPU controlled by each macroeconomic factor on TSM returns.

We observe that Chinese EPU index levels significantly negatively impact TSM strategies' momentum returns unlike RSM momentum returns. However, the results are robust in that we obtained the same directions of signs on each control variable from TSM regressions and RSM regressions.

**Table 10.** Effect of EPU level on return signal momentum strategy return in the Chinese stock market, controlling each of the macroeconomic factors. *** denotes significant at 0.01 level, ** denotes 0.05 significance level, and * represents significant at 0.1 level, respectively.

| | RSM | | | | | | | | | | | | | | | |
|---|---|---|---|---|---|---|---|---|---|---|---|---|---|---|---|---|
| | $TERM_{t-1}$ | | $STIR_{t-1}$ | | $MKT_{t-1}$ | | $MS_{t-1}$ | | $UEMP_{t-1}$ | | $DIV_{t-1}$ | | $IPG_{t-1}$ | | $INF_{t-1}$ | |
| | (1) | (2) | (1) | (2) | (1) | (2) | (1) | (2) | (1) | (2) | (1) | (2) | (1) | (2) | (1) | (2) |
| $EPU_{t-1}$ | −0.033 ** (−2.087) | −0.009 * (−1.662) | −0.024 (−1.633) | −0.006 (−1.616) | −0.015 * (−1.721) | −0.002 (−0.427) | −0.026 * (−1.738) | −0.008 * (−1.751) | −0.024 (−1.649) | −0.007 * (−1.915) | −0.024 (−1.639) | −0.007 * (−1.847) | −0.023 (−1.605) | −0.007 (−0.914) | −0.024 (−1.625) | −0.006 (−1.525) |
| HML | | 0.049 (0.574) | | 0.052 (0.457) | | 0.083 (1.258) | | 0.050 (0.618) | | 0.050 (0.694) | | 0.055 (0.783) | | 0.058 (0.943) | | 0.049 (0.676) |
| RMRF | | 0.209 ** (2.553) | | 0.210 *** (2.996) | | 0.783 *** (6.546) | | 0.212 *** (2.835) | | 0.213 *** (3.132) | | 0.210 *** (3.038) | | 0.224 *** (3.584) | | 0.213 *** (2.661) |
| SMB | | 0.215 *** (4.055) | | 0.230 *** (4.402) | | 0.017 (0.195) | | 0.217 *** (4.538) | | 0.219 *** (4.693) | | 0.224 *** (3.956) | | 0.224 *** (4.893) | | 0.222 *** (4.641) |
| $TERM_{t-1}$ | −0.954 * (−1.833) | −0.207 (−0.768) | | | | | | | | | | | | | | |
| $STIR_{t-1}$ | | | 0.001 (0.026) | 0.016 ** (2.170) | | | | | | | | | | | | |
| $MKT_{t-1}$ | | | | | 0.017 ** (2.482) | 0.053 *** (4.273) | | | | | | | | | | |
| $MS_{t-1}$ | | | | | | | −0.547 (−1.072) | −0.131 (−0.514) | | | | | | | | |
| $UEMP_{t-1}$ | | | | | | | | | −0.218 (−0.543) | −0.052 (−0.258) | | | | | | |
| $DIV_{t-1}$ | | | | | | | | | | | −0.031 (−0.455) | −0.016 (−0.564) | | | | |
| $IPG_{t-1}$ | | | | | | | | | | | | | −0.038 *** (−3.099) | −0.034 (−3.191) | | |
| $INF_{t-1}$ | | | | | | | | | | | | | | | −0.030 (−0.206) | −0.174 (−1.602) |

**Table 11.** Effect of EPU level on time series momentum strategy return in the Chinese stock market, controlling each of the macroeconomic factors. *** denotes significant at 0.01 level, ** denotes 0.05 significance level, and * represents significant at 0.1 level, respectively.

| | TSM | | | | | | | | | | | | | | | |
| | $\text{TERM}_{t-1}$ | | $\text{STIR}_{t-1}$ | | $\text{MKT}_{t-1}$ | | $\text{MS}_{t-1}$ | | $\text{UEMP}_{t-1}$ | | $\text{DIV}_{t-1}$ | | $\text{IPG}_{t-1}$ | | $\text{INF}_{t-1}$ | |
| | (1) | (2) | (1) | (2) | (1) | (2) | (1) | (2) | (1) | (2) | (1) | (2) | (1) | (2) | (1) | (2) |
|---|---|---|---|---|---|---|---|---|---|---|---|---|---|---|---|---|
| $\text{EPU}_{t-1}$ | −0.038 ** (−2.100) | −0.031 *** (−2.774) | −0.029 * (−1.830) | −0.023 *** (−3.329) | −0.028 ** (−2.153) | −0.016 ** (−2.254) | −0.033 ** (−2.091) | −0.028 *** (−3.159) | −0.030 * (−1.875) | −0.025 *** (−3.239) | −0.030 * (−1.788) | −0.024 *** (−3.136) | −0.029 * (−1.794) | −0.024 *** (−2.970) | −0.029 * (−1.752) | −0.023 *** (−3.010) |
| HML | | −0.021 (−0.148) | | −0.012 (−0.112) | | 0.037 (0.284) | | −0.032 (−0.264) | | −0.037 (−0.341) | | −0.012 (−0.124) | | −0.005 (−0.043) | | −0.018 (−0.174) |
| RMRF | | 0.053 (0.314) | | 0.059 (0.430) | | 0.971 *** (7.136) | | 0.061 (0.400) | | 0.063 (0.463) | | 0.062 (0.475) | | 0.061 (0.426) | | 0.062 (0.437) |
| SMB | | 0.086 (0.842) | | 0.109 (1.275) | | −0.223 (−1.645) | | 0.079 (0.966) | | 0.081 (1.102) | | 0.099 (1.143) | | 0.105 (1.402) | | 0.104 (1.348) |
| $\text{TERM}_{t-1}$ | −0.828 (−1.417) | −0.585 (−1.366) | | | | | | | | | | | | | | |
| $\text{STIR}_{t-1}$ | | | 0.005 (0.344) | 0.015 (1.234) | | | | | | | | | | | | |
| $\text{MKT}_{t-1}$ | | | | | 0.022 (0.205) | 0.085 *** (5.106) | | | | | | | | | | |
| $\text{MS}_{t-1}$ | | | | | | | −1.042 * (−1.982) | −0.918 ** (−2.276) | | | | | | | | |
| $\text{UEMP}_{t-1}$ | | | | | | | | | −0.624 (−1.271) | −0.606 * (−1.975) | | | | | | |
| $\text{DIV}_{t-1}$ | | | | | | | | | | | 0.003 (0.040) | 0.005 (0.101) | | | | |
| $\text{IPG}_{t-1}$ | | | | | | | | | | | | | −0.047 *** (−4.411) | −0.044 * (−1.948) | | |
| $\text{INF}_{t-1}$ | | | | | | | | | | | | | | | −0.272 (−1.412) | −0.350 * (−1.858) |

## 6. Results of Event Study Analysis

This section represents the event study analysis results of high-EPU events' effect on the Chinese stock market momentum returns. Table 12 shows the results of the event study, and this paper visualizes the events' effects as shown in Figure 2.

**Table 12.** Results of the event study analysis. *** denotes significant at 0.01 level, ** denotes 0.05 significance level, and * represents significant at 0.1 level, respectively. *Note: p*-values in round brackets.

| Event Window | Average Abnormal Returns | | Event Window | Average Abnormal Returns | |
|---|---|---|---|---|---|
| | **RSM** | **TSM** | | **RSM** | **TSM** |
| (−12, 0) | −1.093 (0.306) | −1.370 (0.208) | (0, +7) | −2.354 ** (0.046) | −3.583 *** (0.007) |
| (−11, 0) | −1.181 (0.272) | −1.423 (0.193) | (0, +8) | −2.643 ** (0.030) | −4.908 *** (0.001) |
| (−10, 0) | −1.748 (0.119) | −1.682 (0.131) | (0, +9) | −2.052 * (0.074) | −3.117 ** (0.014) |
| (−9, 0) | −0.522 (0.616) | −1.082 (0.311) | (0, +10) | −1.691 (0.129) | −2.630 ** (0.030) |
| (−8, 0) | −0.857 (0.417) | −1.311 (0.226) | (0, +11) | −1.538 (0.163) | −2.085 * (0.071) |
| (−7, 0) | −0.726 (0.488) | −1.394 (0.201) | (0, +12) | −1.620 (0.144) | −1.680 (0.131) |
| (−6, 0) | −1.320 (0.223) | −2.065 * (0.073) | (0, +13) | −1.646 (0.138) | −1.554 (0.159) |
| (−5, 0) | −2.192 * (0.060) | −4.155 *** (0.003) | (0, +14) | −2.026 * (0.077) | −2.020 * (0.078) |
| (−4, 0) | −3.989 *** (0.004) | −3.917 *** (0.004) | (0, +15) | −2.216 * (0.058) | −1.757 (0.117) |
| (−3, 0) | −3.148 ** (0.014) | −2.665 ** (0.029) | (0, +16) | −2.417 ** (0.042) | −1.371 (0.208) |
| (−2, 0) | −2.554 ** (0.034) | −2.441 ** (0.041) | (0, +17) | −2.415 ** (0.042) | −1.055 (0.323) |
| (−1, 0) | −1.629 (0.142) | −1.519 (0.167) | (0, +18) | −2.727 ** (0.026) | −1.132 (0.290) |
| (0, 0) | −0.637 (0.542) | −0.929 (0.380) | (0, +19) | −2.729 ** (0.026) | −0.988 (0.352) |
| (0, +1) | −0.433 (0.676) | −0.798 (0.448) | (0, +20) | −2.535 ** (0.035) | −1.009 (0.343) |
| (0, +2) | −0.136 (0.895) | −0.469 (0.652) | (0, +21) | −2.435 ** (0.041) | −0.847 (0.422) |
| (0, +3) | −0.577 (0.580) | −0.880 (0.405) | (0, +22) | −2.359 ** (0.046) | −0.859 (0.416) |
| (0, +4) | −1.339 (0.218) | −3.148 *** (0.008) | (0, +23) | −2.105 * (0.068) | −0.727 (0.488) |
| (0, +5) | −1.902 * (0.094) | −2.623 ** (0.031) | (0, +24) | −2.533 ** (0.035) | −1.389 (0.202) |
| (0, +6) | −2.565 ** (0.033) | −4.000 *** (0.004) | | | |

The high-EPU events constantly negatively impact momentum returns without exception for all event windows. The significant negative influences started five months before the events occurred and gradually decreased in the following seven months. The event study analysis presents similar results to previous regression analyses in that three months after the high-EPU events occur, the EPU level will have an insignificant negative impact on stock momentum returns in the Chinese market. Usually, the aftermath of a high-EPU event will last for nine months. In all cases, RSM returns will respond less to high-EPU events shocks than TSM returns.

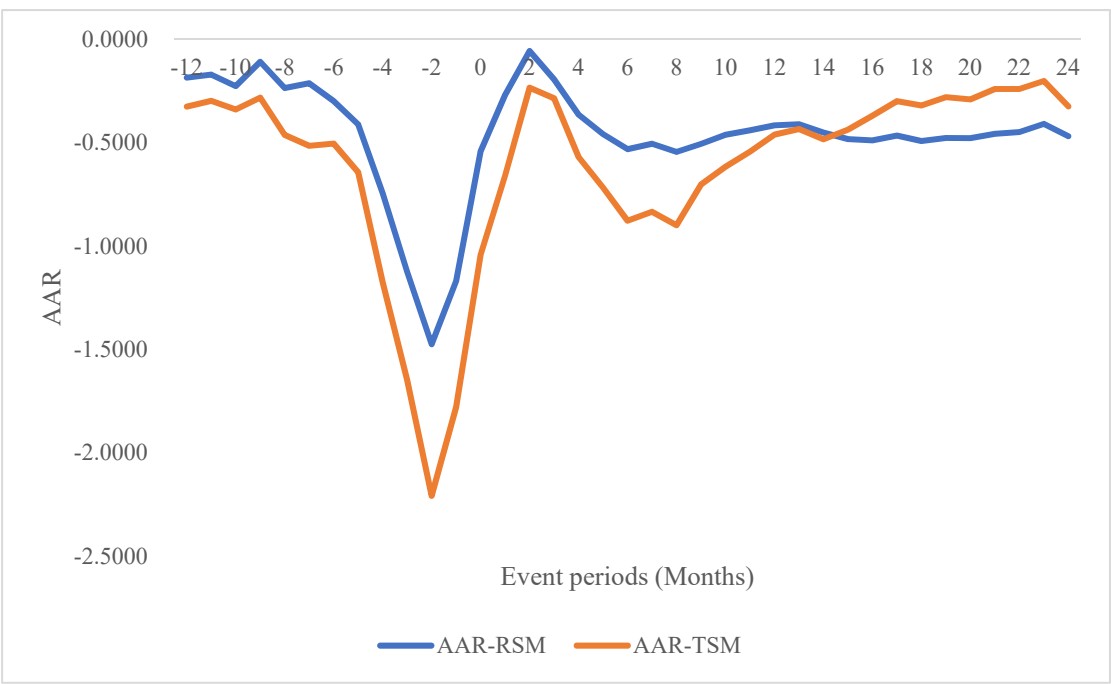

**Figure 2.** Average abnormal return of RSM and TSM. Note: the event periods are from −12 months to 24 months.

## 7. Discussion

The regression analysis used in this study identified the existence of weak momentum effects in the Chinese stock market. These results verify Shi and Zhou's (2017) conclusion that the momentum effects on the Chinese stock market are insignificant. After Fama–French three factor risk adjustments, all static momentum portfolios generate insignificant results. As they are applied to developed markets, investors may hardly obtain considerable profits using conventional fixed threshold momentum strategies in the Chinese market context. However, this paper demonstrates the dynamic threshold for RSM strategies based on EPU and achieves significant outcomes from the risk-adjusted excess returns of EPU-based dynamic RSM strategies, which are about eight times higher than those of static portfolios. The distinction between static and dynamic RSM strategies based on EPU displays that EPU is a critical component that adversely affects momentum returns. EPU-based momentum strategies can help investors make remarkable profits.

Our results corroborate the existing literature that high EPU levels reduce stock market returns (Gong et al. 2022; Luo and Zhang 2020). Our event study method's results support Kundu and Paul's (2022) assertion that the detrimental effects of high-EPU events will diminish in the following two months. The phenomenon implies that the EPU will have a diminishing influence on the Chinese stock market's volatility within two months. However, the existing literature cannot explain why the EPU restarting increases the negative impact on stock market volatility after two months.

Meanwhile, our results partially concur with the literature already available about developed markets, which asserts the negative relationship between EPU and momentum returns by ranking the EPU level of each month within the sample (Brogaard and Detzel 2015; Gu et al. 2021). Our findings indicate that greater investment risks and uncertainties will decrease momentum returns for investors during high-EPU months. However, since investors will be exposed to fewer risks and receive lower returns from the market during low-EPU periods, the EPU level will have no discernible effects on momentum returns. In order to significantly increase their momentum excess returns from TSM and RSM strategies, this paper recommends that investors use momentum strategies during medium-EPU

situations (normal days). This study makes the assumption that appropriate uncertainty will boost Chinese investors' self-confidence and encourage riskier investments.

It is noteworthy to mention that the loser portfolios of all momentum strategies exhibit notably positive responses to Chinese EPU levels. The implication is that momentum returns will suffer a significant negative shock during a high EPU period in the Chinese stock market, because higher EPU level will result in higher loser portfolio excess returns with relatively lower winner portfolio excess returns. According to the 'long winners' and 'short losers' trading rules of momentum strategies, the difference between winner portfolio excess returns and loser portfolio excess returns will get smaller, harming momentum returns. However, our results suggest that this 'differential damage' is weaker in RSM strategies compared with TSM strategies, indicating RSM strategies have superior risk resistance ability than TSM strategies (Papailias et al. 2021). This finding suggests that using RSM strategies in the Chinese stock market will enable investors to have higher resistance to Chinese economic policy uncertainty risks if we ignore the business cycle of the Chinese economy.

Additionally, this paper confirms previous research by showing that EPU will affect momentum returns differently at different stages of business cycles (Paule-Vianez et al. 2020). Surprisingly, a high EPU level will increase the momentum returns during periods when the Chinese economy is expanding. Our research confirms Min and Xiao's (2021) findings that momentum profits will rise significantly at business cycle peaks. We explain this phenomenon as follows: One way that expansion stages will benefit investors is by boosting their confidence (Xu et al. 2021), which will encourage them to keep investing in winner portfolios and boost momentum returns. On the other hand, stock market manipulations will hurt the stock market returns (Cui et al. 2021). However, the stock market manipulations will more frequently occur in China than in developed markets. The high-EPU levels will significantly restrain the happening of the Chinese stock market manipulations and therefore increase the momentum strategies' profitability.

## 8. Conclusions

In our study, we find the weak existence of momentum effects in the Chinese stock market. With a one-standard-deviation of EPU increases, TSM momentum return after risk adjustment will suffer a decrease of 4.5%, while the RSM strategy will decrease by 1.6%. We strongly recommend that investors apply EPU-based dynamic threshold momentum strategies, which yield eight times higher returns than conventional TSM strategies. Investors should be prudent in using momentum strategies during recession cycles with high-EPU levels, which will harm the momentum returns in the Chinese stock market. However, in medium EPU periods (normal days), the Chinese EPU level will significantly boost the momentum returns, and investors can seize such opportunities to gain higher returns. Besides, investors should notice the main drawback of return signal momentum strategies. It needs a relatively long ranking period, usually above six months, to identify the winner and loser portfolios.

Our full sample indicates an insignificant negative impact of EPU on momentum returns by controlling all selected macroeconomic factors. However, the negative impact of EPU will be mitigated in expansion periods compared with recession periods. Furthermore, Huang and Luk's new Chinese EPU index is measured daily, allowing researchers to investigate the short-term (daily frequency) effect of EPU on Chinese stock market momentum returns and develop more profitable strategies.

### 8.1. Policy Recommendations

Apart from recommendations to practitioners, our findings also bring suggestions for policymakers. First, different levels of EPU may result in diverse impacts on the stock returns momentum. During high-EPU stages, policymakers should keep economic policies transparent and stable to mitigate the negative effects on stock returns of uncertainties. Second, the adverse impacts of high-EPU events on momentum returns will be significantly

strengthened two months after the event happens. Although the reason is unclear, policymakers should be aware of such changes and be prepared in advance. Third, the stock market manipulations happen more frequently in China than in developed stock markets. Therefore, we suggest policymakers strengthen the Chinese stock market's supervision to ensure it operates properly.

*8.2. Limitations and Future Recommendations*

In order to investigate the EPU–momentum return relationships in the Chinese stock market through controlling the macroeconomic risk factors, this article is limited by the relevant short time span of available data. Involving a longer period would result in more accurate effects of EPU on momentum returns that provide a reference for investors and policymakers. However, we dealt with this problem by applying multiple research methods and obtained robust results. Moreover, this paper has not examined the performance of short-ranking-periods momentum strategies (i.e., 1–6 months) due to the use of RSM strategies needed above six-month periods to ensure the thresholds can be selected more precisely.

Although the Chinese stock market is regarded as a representative of the emerging markets, we recommend future studies to examine the EPU–momentum returns relationships among various emerging markets globally. The EPU index is available on a daily basis while we merely measured the monthly impact of EPU on momentum returns. Hence, we suggest future research look into the short-term effects of EPU on momentum returns to acquire a comprehensive view of EPU–momentum returns relationships.

**Author Contributions:** Conceptualization, P.Z. and Y.W.; methodology, P.Z.; formal analysis, P.Z.; data curation, Y.W.; writing—original draft preparation, P.Z.; writing—review & editing, P.Z. and Y.W.; visualization, P.Z. All authors have read and agreed to the published version of the manuscript.

**Funding:** This research received no external funding.

**Institutional Review Board Statement:** Not applicable.

**Informed Consent Statement:** Not applicable.

**Data Availability Statement:** The data that support the findings of this study are openly available in the RESSET database. For the new Chinese EPU index used in this study, the data that support the findings of this study are available in Huang and Luk's website at: https://economicpolicyuncertaintyinchina.weebly.com/ (accessed on 7 April 2022).

**Conflicts of Interest:** No potential conflict of interests with respect to the research.

**Appendix A. Steps of the Event Study Method**

The event study analysis adopted as:

1. Determine the value of the intercept term and slope term of OLS regression of the estimate windows.

$$R_t = \alpha + \beta R_{mt} + \varepsilon_t, \ t = T_0 + 1, \cdots, T_1 \quad \text{(A1)}$$

This paper uses CSI 300 index returns as a proxy of market return $R_m$, and $R_t$ is the momentum return at time t.

2. Use the same coefficient $\alpha$ and $\beta$ to estimate the average expected returns during the pre-and post-event windows.

$$ER_t = \alpha + \beta R_{mt} \quad \text{(A2)}$$

where $ER_t$ is the expected normal returns from the event window, $R_{mt}$ is the CSI 300 index returns during the event window.

3. Calculate the abnormal returns.

$$AR_t = R_t - ER_t \quad \text{(A3)}$$

The $AR_t$ denotes the abnormal return during the event period, $R_t$ denotes the actual momentum returns, and $ER_t$ is the expected normal returns from the event window.

4. According to most of the existing literature, this paper tends to investigate the accumulated abnormal returns over a given period. The cumulated abnormal return (CAR) is calculated as Equation (A4):

$$CAR_{t:t+k} = \sum_{k} AR_{t:t+k} \tag{A4}$$

5. For each high EPU event *i*, this paper takes the average value of each period *t: t+k*, and calculates the average cumulated abnormal returns.

$$\overline{ACAR_{t:t+k}^{i}} = \frac{1}{N} \sum_{n=1}^{N_i} CAR_{t:t+k}^{n} \tag{A5}$$

where $N$ denotes the total number of high EPU events, and $\overline{ACAR_{t:t+k}^{i}}$ means the average cumulated abnormal return from $t$ to $t + k$.

6. Establish hypothesis tests to obtain the significance level of the average abnormal return with $H_0$: the average abnormal return $AR$ has a mean of 0.

$$t_{CAR} = \frac{\overline{CAR_{it}} \cdot \sqrt{n}}{\sigma(CAR_{it})} \tag{A6}$$

where the $t_{CAR}$ denotes the test statistics of a specific event's cumulated abnormal return.

## Note

[1] The URL of the website is: https://economicpolicyuncertaintyinchina.weebly.com/ (accessed on 7 April 2022).

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
