# Peer review of "How Does Economic Policy Uncertainty Affect Momentum Returns? Evidence from China"

_ijfs, doi:10.3390/ijfs10030059_

Round 1

Reviewer 1 Report

The reviewed study is interesting and concerns an important issue. The Chinese economy is one of the most important in the world and therefore it is good that Authors have made an analysis the impact of the Chinese EPU levels on stock market momentum returns. This paper has filled the knowledge gap becouse that no existing literature has studied the relationship between EPU and momentum returns in emerging markets. The study has the correct structure and based on a wide literature review. However, it seems that Authors presented the research results in too much detail in the introduction. This may discourage potential readers from reading the rest part of the paper.

Author Response

Dear professor, thanks for your review. We appreciate your valuable comments, which are beneficial for improving our study. We have noticed that we have put too many results in the introduction. Therefore, we deleted points 4 and 5 to let the readers discover the findings in the rest parts of our research.

Reviewer 2 Report

The topic is interesting, and the author has done a great job in realizing the subject. However, there are few areas on the paper that is still lagging and should be addressed properly.

Abstract

1.     The authors should motivate the choice of variables  with theory and empirical backing on the subject

2.     Keywords should be revised to match key element of title

3.     Rewrite the title to be more catchy

4.     Introduction

1.     The objective of the paper presented need more clarifications to suit reader to understand the main idea of the paper especially for the study case is needed

2.     Literature review

 The literature is well written. However, there is need for more recent studies ranging from 2018-2022 to motivate the study properly. The entire study is too scanty and the related literature is not exhausted

Economic Policy Uncertainty and Energy Prices: Empirical Evidence from Multivariate DCC-GARCH Models. Energies15(10), 3712

Methodology

1.     This section is generally well motivated, Kindly take note of the following minor additions

2.     More benefit of the various techniques utilized should be stated

3.     Check for cross-sectional dependency and add correlation text and VIF

4.     The authors should avoid much mathematical expressions or take some to appendix and make the study reader friendly for other practitioner other than academic with out compromise for study intend and quality.

Discussion

1.     The discussion is well written, but the authors should like their findings to the previous studies in the literature.

2.     There is need for professional proofreading or consult English native support

3.     Conclusion

1.     The sub-title should be conclusion and policy recommendation but not only conclusion

2.     The policy which is the engine of the study is weak and small. I therefore encourage the authors to elaborate more on the policy recommendations to policy makers for the investigated bloc

3.     The authors should add limitation of the study and future recommendation

Author Response

Dear professor, many thanks for your comments on our manuscript. We have carefully checked every points and those comments are all valuable and very helpful for revising and improving our paper.

The detailed revisions are listed in the attached MS Word file.
